# A Bayesian approach to single-particle electron cryo-tomography in RELION-4.0

**Jasenko Zivanov[1,2,3], Joaquín Otón[1,4], Zunlong Ke[1,5], Andriko von Kügelgen[1,6], Euan Pyle[7], Kun Qu[1], Dustin Morado[1,5], Daniel Castaño-Díez[3,8], Giulia Zanetti[7], Tanmay AM Bharat[1,6], John AG Briggs[1,5]\*, Sjors HW Scheres[1]\***

[1]MRC Laboratory of Molecular Biology, Cambridge, United Kingdom; [2]Laboratory of Biomedical Imaging (LIB), Lausanne, Switzerland; [3]BioEM lab, Biozentrum, University of Basel, Basel, Switzerland; [4]ALBA Synchrotron, Barcelona, Spain; [5]Max Planck Institute of Biochemistry, Martinsried, Germany; [6]Sir William Dunn School of Pathology, University of Oxford, Oxford, United Kingdom; [7]Institute of Structural and Molecular Biology, Birkbeck College, London, United Kingdom; [8]Instituto Biofisika, Leioa, Spain

**\*For correspondence:**
briggs@biochem.mpg.de (JAGB);
scheres@mrc-lmb.cam.ac.uk (SHWS)

**Abstract** We present a new approach for macromolecular structure determination from multiple particles in electron cryo-tomography (cryo-ET) data sets. Whereas existing subtomogram averaging approaches are based on 3D data models, we propose to optimise a regularised likelihood target that approximates a function of the 2D experimental images. In addition, analogous to Bayesian polishing and contrast transfer function (CTF) refinement in single-particle analysis, we describe the approaches that exploit the increased signal-to-noise ratio in the averaged structure to optimise tilt-series alignments, beam-induced motions of the particles throughout the tilt-series acquisition, defoci of the individual particles, as well as higher-order optical aberrations of the microscope. Implementation of our approaches in the open-source software package RELION aims to facilitate their general use, particularly for those researchers who are already familiar with its single-particle analysis tools. We illustrate for three applications that our approaches allow structure determination from cryo-ET data to resolutions sufficient for de novo atomic modelling.

## Editor's evaluation

Single-particle tomography (SPT) is a useful method to determine the structure of proteins imaged in situ. This important work presents an easy-to-use tool for SPT that approximates the use of 2D tomographic projections using a 'pseudo-subtomogram' data structure, chosen to facilitate implementation within the existing RELION codebase. The examples shown provide solid support for the claims about the efficacy of the approach.

## Introduction

In recent years, electron cryo-microscopy (cryo-EM) has allowed the 3D imaging of an increasing number of biological macromolecules at resolutions sufficient for de novo atomic modelling. This development, originally driven by advances in detector technology, was further facilitated by novel, robust image processing algorithms. In single-particle analysis, images of multiple copies of isolated macromolecular complexes, or particles, that are suspended in random orientations in a thin layer of vitreous water are combined in a 3D reconstruction. Nowadays, many aspects of single-particle analysis workflows can be performed with only minimal human supervision, for example, the detection, extraction, and initial classification of particles in the images (*Zivanov et al., 2018*; *Bepler et al.,*

*2019*; *Wagner et al., 2019*), 3D reconstruction (*Zivanov et al., 2018*; *Punjani et al., 2017*), as well as refinement of the optical parameters (*Zivanov et al., 2018*; *Zivanov et al., 2020*; *Punjani et al., 2017*; *Tegunov et al., 2021*) and per-particle tracking of electron beam-induced motion (*Zheng et al., 2017*; *Zivanov et al., 2019*). Many of the algorithms that underlie these modern methods are built on solid statistical foundations that require few tunable parameters. This decreases the need for operator expertise and provides objectivity, as well as robustness, in obtaining optimal structures.

The single-particle approach is, however, limited to investigating isolated protein complexes that are purified to relative homogeneity. To examine these complexes in their crowded physiological environment, electron cryo-tomography (cryo-ET) may be used instead. In the tomographic approach, the sample is tilted multiple times during image acquisition, yielding a so-called tilt series of images from which a 3D tomogram can be computed. In the same manner as single-particle analysis, repeated occurrences of particles in those tomograms can then be aligned and averaged to obtain higher-resolution reconstructions. This process is referred to as *subtomogram averaging*. Unlike the field of single-particle analysis, labs use many different tools for subtomogram averaging (e.g. *Kremer et al., 1996*; *Nickell et al., 2005*; *Heumann et al., 2011*; *Castaño-Díez et al., 2012*; *Hrabe et al., 2012*; *Chen et al., 2013*; *Galaz-Montoya et al., 2016*; *Sanchez et al., 2019*; *Chen et al., 2019*; *Jiménez de la Morena et al., 2022*) and many of the tools used require considerable levels of expertise from the operator, often in order to tune parameters that arise from heuristics in the underlying algorithms. This not only provides a barrier for new scientists entering the field, but can also lead to the calculation of suboptimal structures.

Compared to single-particle analysis, subtomogram averaging faces several unique challenges. In addition to estimating the position and orientation of each particle, the algorithm also has to consider the geometry of the tilt series. Typically, this is solved through a set of preprocessing steps that include estimation of contrast transfer function (CTF) parameters and alignment of the tilt series, followed by the reconstruction of, often inconveniently large, tomograms for the entire field of view. Smaller subtomograms, centred around selected particles, are then extracted from the tomograms and used in a separate process of subtomogram alignment and averaging. The separation between tomogram reconstruction and subtomogram averaging can lead to an accumulation of errors, because errors in the CTF estimation or tilt-series alignments are hard to correct. In addition, because the sample cannot be rotated 180° within the microscope, the subtomograms contain empty regions in Fourier space, the so-called missing wedge, which are difficult to deal with in subtomogram averaging (e.g. see *Schmid and Booth, 2008*; *Förster et al., 2008*; *Bartesaghi et al., 2008*; *Frank et al., 2012*).

A fundamental problem with subtomogram averaging as described above is that it transforms the original 2D image data into 3D subtomograms, which are then used as a substitute for experimental data in the alignment algorithm. RELION-2 introduced the concept of a 3D CTF to describe the transfer of information from the 2D images to the subtomograms, which dealt to some extent with the missing wedge and the loss of information through interpolations in the reconstruction algorithm (*Bharat and Scheres, 2016*). A drawback of the 3D CTF approach is that it does not deal correctly with the lower resolution regions of Fourier space, where information from different tilt images overlaps. A statistically more attractive approach would be to formulate the optimisation target function directly as a function of the actual 2D images that are measured in the microscope. This has been proposed in an approach called constrained single-particle cryo-ET (*Bartesaghi et al., 2012*), where individually boxed particles from the tilt-series images are processed as in single-particle analysis, but their relative orientations are kept fixed. A similar approach was also implemented in the program emClarity (*Himes and Zhang, 2018*). To deal with unknowns in the relative orientations of the particles from the tilt-series images, as well as their CTFs, the program M recently introduced new optimisation approaches that compare reference projections against the 2D particle images (*Tegunov et al., 2021*). M relies on RELION for alignment and classification of 3D subtomograms that are recalculated from the optimised parameters in M. Nevertheless, this iterative approach allows subtomogram averaging to resolutions that approach those observed for single-particle analysis, even for particles in complex cellular environments (*Tegunov et al., 2021*).

Here, we describe a new approach to subtomogram averaging in RELION-4.0 that optimises a regularised likelihood function that approximates the direct use of the 2D images of the tilt series. In order to do so at acceptable computational and implementation costs, we have altered the main refinement program in RELION-4.0 to work with so-called *pseudo-subtomograms*: explicitly

constructed sets of 3D data arrays that contain sums of CTF pre-multiplied 2D tilt-series images, together with auxiliary arrays that contain the corresponding sum of squared CTFs and how often each 3D voxel has been observed. Pseudo-subtomograms no longer aim to represent the actual scattering potential of the underlying particles, in the way that conventional subtomograms would. Instead, they represent a convenient way to implement an approximation to the 2D approach within the existing RELION code. Evaluation of the pseudo-subtomograms by RELION-4.0 approximates the likelihood of observing a hypothetical particle in the images of the entire tilt series, given the current model. Using that likelihood as a metric, operations equivalent to those in single-particle analysis can now be performed on tomographic data, for example, 3D initial model generation, 3D classification, or high-resolution refinement. In addition, we describe new methods for optimising parameters of the tilt series that exploit the increased signal-to-noise ratio in the average structure. Besides optimisation of the tilt-series alignment itself, we also describe methods analogous to CTF refinement (*Zivanov et al., 2018*; *Zivanov et al., 2020*) for refining descriptors of the optical properties (defocus, astigmatism, and higher-order aberrations) and a method akin to Bayesian polishing (*Zivanov et al., 2019*) to model beam-induced particle motion throughout the tilt series. Once all these parameters have been optimised, new pseudo-subtomograms can be constructed and the alignment can be repeated. The resulting iterative image processing workflow is similar to existing approaches for single-particle analysis in RELION.

## Methods
### Particle alignment and averaging

RELION performs maximum a posteriori estimation to find the set of model parameters $\Theta$ that maximise the probability of observing the experimental images $\mathbb{X}$. Using Bayes' theorem, we define a regularised likelihood optimisation target function as

$$P(\Theta|\mathbb{X}) = P(\mathbb{X}|\Theta)P(\Theta),  \tag{1}$$

where $P(\Theta)$ expresses prior information about the model, that is, that the reconstructed map has limited power in Fourier space, and $P(\mathbb{X}|\Theta)$ is the likelihood of observing the data given the model. A marginalised likelihood function is used, where one integrates over the unknown alignments $\phi$ of each individual particle. For simplicity, these integrals are omitted from the notations used in this article.

The data model assumes independent Gaussian noise on the Fourier components of the cryo-EM images of individual particles $p$. We therefore write the negative log-likelihood of observing a particle $p$ in a hypothetical alignment $\phi$ as a sum over a grid of 2D Fourier pixels $\mathbf{j} \in \mathbb{R}^2$:

$$-\log\left(P(X^p|\phi)\right) \propto \sum_{\mathbf{j}} \frac{|X_{\mathbf{j}}^p - \mathrm{CTF}^p(\mathbf{j})V_{\mathbf{j}}^{(p)}|^2}{\sigma_j^2},  \tag{2}$$

where $X^p$ is the Fourier transform of the experimental particle image, $\mathrm{CTF}^p$ its contrast-transfer function, $V_{\mathbf{j}}^{(p)}$ denotes the 2D slice out of the 3D Fourier transform of the known map $V$ into the view of the particle, and $\sigma_j^2$ is the noise variance of the frequency band of $\mathbf{j}$ given by

$$V_{\mathbf{j}}^{(p)} = \exp(i\mathbf{t}_p \cdot \mathbf{j})V(A_p\mathbf{j})  \tag{3}$$

for a 2D vector $\mathbf{t}_p$ and a $2 \times 3$ matrix $A_p$ that respectively encapsulate the particle's position and orientation, and the evaluation of $V(A_p\mathbf{j})$ is achieved through linear interpolation.

In tomography, our aim is to approximate that same likelihood on tilt-series data. The equivalent is a sum over the pixels of the relevant regions of all images $f$ from the tilt series:

$$-\log\left(P(X^p|\phi)\right) \propto \sum_{f,\mathbf{j}} \frac{|X_{f\mathbf{j}}^p - \mathrm{CTF}_f^p(\mathbf{j})V_{f\mathbf{j}}^{(p)}|^2}{\sigma_j^2}.  \tag{4}$$

We model the shifts and rotations as compositions of per-particle and per-image components:

$$\mathbf{t}_{pf} = A_f^\mathsf{T}\mathbf{T}_{pf} + \mathbf{t}_f  \tag{5}$$

$$A_{pf} = R_p A_f, \tag{6}$$

where we keep the per-particle rotation component, $R_p$, identical for all images in the tilt series, and only vary $A_f$, the rotational alignment of the tilt-series images. In turn, the tilt-series alignment $A_f$ is shared among all particles in a given tilt image. The per-particle part of the translation is modelled as a 3D vector, $\mathbf{T}_{pf} \in \mathbb{R}^3$, that can vary over different tilt images $f$. This contrasts with single-particle analysis, where beam-induced motion of the particle can be corrected for as a preprocessing step (*Li et al., 2013*; *Scheres, 2014*; *Zheng et al., 2017*; *Zivanov et al., 2019*), so that each particle is associated with a single 2D translation in a motion-corrected image.

For our pseudo-subtomogram approach, we now approximate the sum over 2D pixels $\mathbf{j}$ and tilt images $f$ in *Equation 4* by a sum over 3D voxels $\mathbf{k}$ in the pseudo-subtomogram:

$$-\log\left(P(X|\phi)\right) \propto \sum_{\mathbf{k}} \frac{|D_{\mathbf{k}}^p - W_{\mathbf{k}}^p V(R_p \mathbf{k})|^2}{M^p \sigma_{\mathbf{k}}^2}. \tag{7}$$

Here, the data term $D^p$, the weight term $W^p$, and the multiplicity term $M^p$ are 3D arrays in the Fourier domain. Together, they constitute a pseudo-subtomogram. They are constructed as follows:

$$D_{\mathbf{k}}^p = \sum_{f,\mathbf{j}} l(A_{pf}\mathbf{j} - \mathbf{k}) \mathrm{CTF}_f^p(\mathbf{j}) X_{f\mathbf{j}}^p \tag{8}$$

$$W_{\mathbf{k}}^p = \sum_{f,\mathbf{j}} l(A_{pf}\mathbf{j} - \mathbf{k}) |\mathrm{CTF}_f^p(\mathbf{j})|^2 \tag{9}$$

$$M_{\mathbf{k}}^p = \sum_{f,\mathbf{j}} l(A_{pf}\mathbf{j} - \mathbf{k}), \tag{10}$$

where $l(\cdot)$ represents linear interpolation with forward mapping, that is, each 2D Fourier pixel $\mathbf{j}$ is projected into 3D Fourier space, updating the eight closest voxels.

Ignoring the difference of pre-multiplying the images with their CTFs, *Equation 7* aims to be equivalent of *Equation 4*. The variance $\sigma_{\mathbf{k}}^2$ is equivalent to $\sigma_j^2$, the power of the noise in the individual Fourier components in the 2D images.

We then optimise *Equation 1* by expectation-maximisation (*Dempster et al., 1977*), using *Equation 7* to construct the likelihood function and using a prior $P(\Theta) \propto \exp\sum_{\mathbf{k}} \frac{|V_{\mathbf{k}}|^2}{-2\tau_k^2}$, based on the expected frequency-dependent power of the signal $\tau_k^2$. This leads to the following iterative algorithm:

$$V_k^{(n+1)} = \frac{\sum_p D(R_p^{\mathsf{T}}\mathbf{k})/\sigma_{\mathbf{k}}^{2(n+1)}}{\sum_p W(R_p^{\mathsf{T}}\mathbf{k})/\sigma_k^{2(n+1)} + 1/\tau_k^{2(n+1)}} \tag{11}$$

$$\sigma_k^{2(n+1)} = \frac{\sum_p \sum_{\mathbf{k} \in S_k} |D_{\mathbf{k}} - W_{\mathbf{k}} V(R_p \mathbf{k})|^2}{2 \sum_p \sum_{\mathbf{k} \in S_k} M_{\mathbf{k}}}, \tag{12}$$

$$\tau_k^{2(n+1)} = \frac{|V_{\mathbf{k}}|^2}{2} \frac{\sum_p \sum_{\mathbf{k} \in S_k} W_{\mathbf{k}}}{\sum_p \sum_{\mathbf{k} \in S_k} M_{\mathbf{k}}} \tag{13}$$

where (n) denotes the iteration; the divisions by $\tau_k^2$ and $\sigma_k^2$ in *Equation 11* are evaluated element-wise; and $\tau_k^2$ and $\sigma_k^2$ are calculated by averaging over $\tau_{\mathbf{k}}^2$ and $\sigma_{\mathbf{k}}^2$, respectively, in hollow spheres of radius $k$ and thickness 1, described by $S_k$. The ratio of the terms containing $W_{\mathbf{k}}$ and $M_{\mathbf{k}}$ in *Equation 13* corrects the estimate for the power of the signal from the CTF-corrected map $V$ by the average CTF[2] to account for the fact that the likelihood in *Equation 7* was calculated for CTF pre-multiplied images.

## Pre-oriented priors

Many proteins are organised in imperfect 2D arrays inside the tomograms, for example, inside membranes or as part of capsid-like structures. Often, the individual protein molecules inside these arrays exhibit limited rotational freedom with respect to the surface normal of the array, although they may be able to rotate freely around that normal. This knowledge is often exploited in subtomogram averaging approaches through local orientational searches, for example, see *Förster et al., 2005*. This not only accelerates the refinement, as fewer orientations need to be evaluated, it also makes it possible

to solve more challenging structures because fewer solutions are allowed. In RELION, local orientational searches are implemented as Gaussian priors on the Cartesian translations and on the three Euler angles that describe rotations (*Scheres, 2012*). One advantage of using pseudo-subtomogram alignment is that the coordinate system of the pseudo-subtomograms themselves can be chosen arbitrarily. By default, pseudo-subtomograms are created in the same orientation as the tomogram, but the user can choose to orient them in a more meaningful way. For example, by constructing the pseudo-subtomograms with their $Z$-axis parallel to the 2D array, using a rotational prior of approximately 90° on the tilt angle will limit the amount of rocking of the particles inside the array, while avoiding singularities in the definition of the Euler angles that occur when the tilt angle is close to 0°.

## Tilt-series refinement

Averaging over multiple particles leads to an increased signal-to-noise ratio in the estimated density map $V$. We also implemented procedures that exploit $V$ for subsequent re-estimation of parameters that describe the tilt series. These procedures do not require pseudo-subtomograms, but are performed by comparing projections of the density maps directly with the (Fourier) pixels of 2D boxes that are extracted from the tilt-series images, with a sufficient size to hold the CTF-delocalised signal. The various tilt-series parameters are then estimated by minimising the negative log-likelihood as defined in *Equation 4*, that is, the sum over noise-weighted square differences between the prediction and the observation.

The tilt-series properties that can be refined fall into two broad categories: optical and geometrical. The optical refinement concerns the different parameters of the CTF, while the geometrical refinement aims to optimise the alignment of the tilt series, as well as the beam-induced motion of the individual particles. Both sets of algorithms are closely related to the corresponding single-particle algorithms in RELION: optical-aberration refinement (*Zivanov et al., 2018*; *Zivanov et al., 2020*) and Bayesian polishing (*Zivanov et al., 2019*), respectively. In spite of the similarity between the algorithms, the models that are optimised differ significantly from single-particle analysis. Details of the implementation of the optical and geometrical refinement algorithms are given in Appendix 1. We also note that Bayesian polishing in SPA describes particle motions between individual movie frames. Although our approach for tomography can also consider movie frames, the current implementation uses the same regularisation of particle motions between movie frames within each tilt image as between the movie frames from other tilt images. Because preliminary tests showed limited benefits in considering the movie frames in this manner, only the functionality to model particle motions between the tilt-series images was exposed on the GUI.

CTF refinement for tomographic data in RELION-4.0 includes optimisation of scale factors that model frequency-dependent radiation damage, defocus, astigmatism, and higher-order symmetrical and antisymmetrical aberrations. Although individual particles within a field of view are at distinct defoci in the tilt-series images, their relative defoci are known from the geometry of the tilt series and the known 3D positions of the particles in the tomogram. Therefore, one can efficiently perform defocus estimation in a single pass, considering all particles in a tilt-series image simultaneously. In order to do so, we modified procedures that were developed for higher-order aberration estimation in single-particle analysis (*Zivanov et al., 2020*), where the information from all particles in each tilt-series image is condensed into two images that are used to estimate a common phase shift (see Appendix).

Similar procedures can also be used to model higher-order symmetrical and antisymmetrical aberrations in the tomographic data. Analogously to our single-particle approach, they are modelled using Zernike polynomials and estimated in the same way. Because the higher-order aberrations are often only a limiting factor at relatively high spatial frequencies, a large number of particles are needed to estimate them reliably. Optimally, higher-order aberrations would thus be estimated globally, over the entire data set, and only for cases that yield high-resolution averages. If aberrations change during data collection, data sets may be split into optics groups, for which aberrations are estimated separately. Typically, the third-order antisymmetrical aberrations are the most important ones, that is, trefoil and axial coma, which can both be caused by a tilted beam. The resolution gains that these optimisations will yield depend on the microscope (mis)alignment. Provided alignment has been performed reasonably well, higher-order aberration correction will probably be most useful for reconstructions that extend beyond 3 Å resolution.

The geometric alignment includes both (rigid) rotational and translational re-alignment of the tilt-series images, as well as the modelling of beam-induced motion of individual particles throughout the tilt series. For the latter, we neglect rotations of the particles, and only model beam-induced translations. By doing so, we can precompute the likelihood of each particle being in each position around its original one, and then look for an alignment that simultaneously maximises the sum of those likelihoods over all tilt-series images and all particles, as well as a prior that ensures spatially coherent motion. This allows us to evaluate the likelihood of a hypothetical particle position by looking up a single interpolated value in an image. In this formulation, the problem becomes equivalent to the Bayesian polishing approach that we originally developed for single-particle analysis, except for the inclusion of a third spatial dimension for the motion.

## Results

We tested our approach on three test cases. *Appendix 2—table 1* provides experimental details for each of the data sets; *Appendix 2—table 2* provides details on the image processing.

### HIV-1 immature capsid

We tested the workflow above on the cryo-ET data set that was used to determine the structure of the immature capsid lattice and spacer peptide 1 (CA-SP1) regions of the Gag polyprotein from human immunodeficiency virus 1 (HIV-1) (*Schur et al., 2016*) (EMPIAR-10164). We used the same subset of five tomograms that were also used to assess the NovaCTF (*Turoňová et al., 2017*), emClarity (*Himes and Zhang, 2018*), and Warp (*Tegunov and Cramer, 2019*) programs. Introducing 3D CTF correction, and using the alignment parameters from the original analysis by Schur et al., NovaCTF reported a resolution of 3.9 Å (*Turoňová et al., 2017*). The Warp program introduced local and global motion correction in the tilt-series images, as well as optimisation of CTF parameters. The combination of Warp and subtomogram alignment and averaging in RELION-3 led to a resolution of 3.8 Å. A recent application of emClarity led to a reconstruction to 3.3 Å resolution (*Ni et al., 2022*).

We used tilt-series projections after movie frame alignment from the original analysis (*Schur et al., 2016*), without any other preprocessing step, along with the tilt-series alignment data, performed with IMOD package (*Kremer et al., 1996*), and CTF parameters estimation using CTFFIND4 (*Rohou and Grigorieff, 2015*). We used 12,910 particles from the five tomograms subset, reconstructed an initial

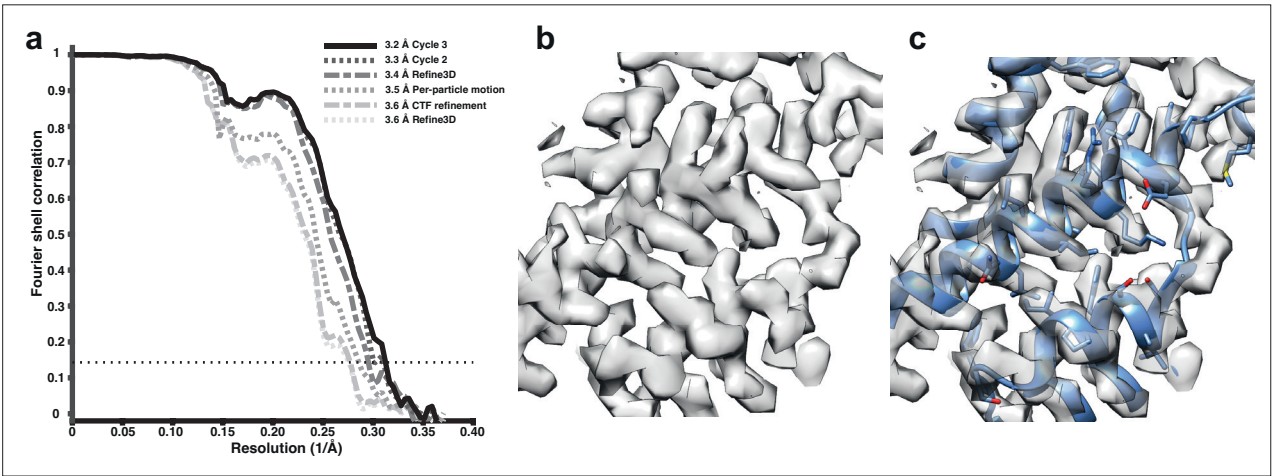

**Figure 1.** Subtomogram averaging of the HIV-1 immature capsid. (**a**) Fourier Shell Correlation (FSC) for resolution estimation of iteratively improved reconstructions using the new RELION-4.0 workflow. (**b**) Representative region of reconstructed density in the final map. (**c**) The same density as in (**b**), together with the published atomic model 5L93, which has not been additionally refined in the density.

The online version of this article includes the following figure supplement(s) for figure 1:

**Figure supplement 1.** Iterative map improvement.

**Figure supplement 2.** Comparison with emClarity.

**Figure supplement 3.** Comparison with M/RELION-3.1.

reference map using the original published particle alignment, and filtered it to 5 Å. A first alignment in 3D auto-refine, followed by averaging of the initial pseudo-subtomograms, led to a resolution of 3.6 Å. This average was then used for a full cycle of pseudo-subtomogram improvement and realignment. We first applied CTF refinement to optimise the defoci of all particles. This improved the resolution only marginally. Subsequent optimisation of the tilt-series geometry, including modelling local particle motion, improved the resolution to 3.5 Å. Finally, realignment of newly generated pseudo-subtomograms led to a resolution of 3.4 Å. A second cycle of these three steps provided 3.3 Å, while a third cycle converged to 3.2 Å (*Figure 1a*). Geometrical refinement was performed estimating local particle motion. The consideration of deformations did not show additional improvement. In the first cycle, where improvements in both CTFs and geometry are most obvious, the order of applying those optimisations did not alter the final result for this data set. These data and results are also distributed as part of the subtomogram tutorial in RELION-4.0, as described on https://relion.readthedocs.io/en/release-4.0/. *Figure 1—figure supplement 1* shows the improvement in map quality during the iterative refinement process; *Figure 1—figure supplement 2* shows a comparison with the 3.3 Å map from emClarity.

Analysis of the complete data set generated a structure at 3.0 Å resolution (*Figure 1—figure supplement 1*), which is the same resolution obtained using the M and RELION-3 workflow (*Tegunov et al., 2021*; *Figure 1—figure supplement 3*), and is likely limited by flexibility and asymmetry in the CA hexamer.

### *Caulobacter crescentus* S-layer

We also applied our approach to thin cellular appendages of *C. crescentus* bacteria known as stalks, which have previously been imaged using cryo-ET (*Bharat et al., 2017*). The cell body and cell stalks of *C. crescentus* cells are covered by a nearly hexagonal, paracrystalline array known as the surface layer (S-layer) (*Smit et al., 1992*). The structure of the S-layer was solved using a combination of X-ray crystallography, cryo-EM single-particle analysis, and subtomogram averaging, revealing how the S-layer

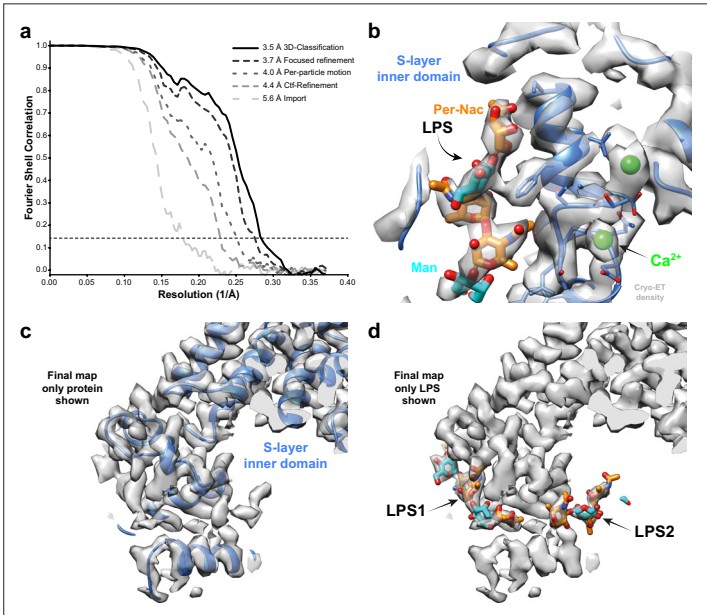

**Figure 2.** Subtomogram averaging of the *C. crescentus* S-layer from cell stalks. (**a**) FSC for resolution estimation of iteratively improved reconstructions using the new RELION-4.0 workflow, tested on the S-layer inner domain. (**b**) Densities for the previously identified lipopolysaccharide (LPS) (cyan and orange) and $Ca^{2+}$ ions (green) in prior electron cryo-microscopy (cryo-EM) single-particle analyses are resolved. (**c, d**) The final map shows two densities for bound LPS O-antigen chains. Panel (**c**) shows only the S-layer protein as blue ribbon and (**d**) shows LPS O-antigen as orange and cyan sugars corresponding to the N-acetyl-perosamine and mannose moieties, respectively.

The online version of this article includes the following figure supplement(s) for figure 2:

**Figure supplement 1.** Comparison of subtomogram averaging (STA) and single-particle analysis (SPA) reconstructions of the *C. crescentus* RsaA.

is attached to bacterial cells by an abundant glycolipid called lipopolysaccharide (LPS) (*Bharat et al., 2017*; *von Kügelgen et al., 2020*). Previously, cryo-ET of the S-layer, using 110 tilt series collected with a dose-symmetric scheme, yielded 51,866 hexamers of the S-layer. This study used a subtomogram averaging approach that is based on a constrained cross-correlation approach implemented in the AV3 MATLAB suite (*Förster and Hegerl, 2007*), and which was specifically optimised for the analysis of macromolecules arranged in a lattice (*Wan et al., 2017*). A 7.4 Å reconstruction of the S-layer was obtained, in which alpha-helices were resolved (*Bharat et al., 2017*). This reconstruction was improved by application of NovaCTF (*Turoňová et al., 2017*), leading to a 4.8 Å reconstruction, in which large amino acid residue side chains were resolved (*von Kügelgen et al., 2020*). Moreover, density for an LPS molecule was observed near the putative LPS-binding residues of the S-layer, in agreement with a cryo-EM single-particle structure of an in vitro reconstituted coplex (*von Kügelgen et al., 2020*). We used the tilt series after movie frame alignment from the initial analysis (*Bharat et al., 2017*), along with the tilt-series alignments performed within IMOD (*Kremer et al., 1996*), CTF parameters from CTFFIND4 (*Rohou and Grigorieff, 2015*), and the Euler angle assignments and subtomogram coordinates from the original analysis. These parameters were imported into RELION-4.0, followed by multiple cycles of pseudo-subtomogram generation and refinement, analogous to the immature HIV-1 data set described above, leading to a 5.6 Å reconstruction of the S-layer hexamer (*Figure 2a*). Next, we defined a mask around the central pore of the S-layer, corresponding to the inner domain bound to LPS, to perform focused refinements. Another cycle of pseudo-subtomogram reconstruction, CTF refinement, and refinement within the new mask improved the resolution to 4.4 Å. Accounting for per-particle motions with additional cycles of pseudo-subtomogram improvements and refinements increased the resolution of the central pore to 4.0 Å, and the inner domain of the S-layer to 3.7 Å. Further 3D classification without alignments identified a subset of 42,990 subtomograms that gave a 3.5 Å resolution reconstruction of the inner S-layer.

The 3.5 Å map is in excellent agreement with the single-particle structure of the in vitro reconstituted complex, including the LPS binding site (*von Kügelgen et al., 2020*; see *Figure 2—figure supplement 1*). Furthermore, divalent metal ions, known to be tightly bound to the inner S-layer (*Matthew, 2021*), are resolved (*Figure 2b*). Surprisingly, at lower isosurface contour levels, we also observed a second LPS binding site (*Figure 2c and d*). The size and shape of this density agree with the structure of the LPS O-antigen, illustrating how improved subtomogram averaging in RELION-4.0 can help uncover new biology.

## Coat protein complex II

Finally, we applied our approach to the *Saccharomyces cerevisiae* coat protein complex II (COPII), which mediates the transport of newly synthesised proteins from the endoplasmic reticulum (ER) to the Golgi apparatus as part of the secretory pathway. COPII is formed by five proteins that assemble sequentially on the ER membrane to induce remodelling of the bilayer into coated carriers in a process known as COPII budding, while simultaneously selectively recruiting cargo into these budding membrane carriers. COPII budding can be reconstituted in vitro from purified proteins and artificial membranes, to form small, spherical vesicles, or long, straight tubes. Cryo-ET has previously been used to visualise the architecture of COPII on reconstituted tubules (*Hutchings et al., 2018*; *Hutchings et al., 2021*). The coat assembles into two concentric layers; the inner layer forms a pseudo helical lattice, which has previously been solved to 4.6 Å resolution using Dynamo-based subtomogram averaging protocols (*Castaño-Díez et al., 2012*).

We used the tilt series after movie alignment from the initial analysis (*Hutchings et al., 2021*), along with the tilt-series alignments performed in Dyname (*Castaño-Díez et al., 2012*) and CTF parameters from CTFFIND4 (*Rohou and Grigorieff, 2015*). COPII-coated tubes were manually traced in the resulting tomograms, and particles were extracted by randomly oversampling their surface, with approximate initial orientations assigned based on the cylindrical geometry. Dynamo was used for initial alignment of 8× binned subtomograms to define the centre of the particles and the directionality of individual tubes. We then imported the particle coordinates for processing in RELION-4.0 using 3D refinement at 4× and 2× binning. Since we expect inner coat subunits to arrange in a lattice, we cleaned the data set by excluding any subtomograms that did not conform to the expected geometrical relationship with their neighbouring particles. A first 3D refinement of the unbinned data set gave a map at 4.4 Å resolution, which was further improved to 4.2 Å and 4.0 Å by tilt-series frame

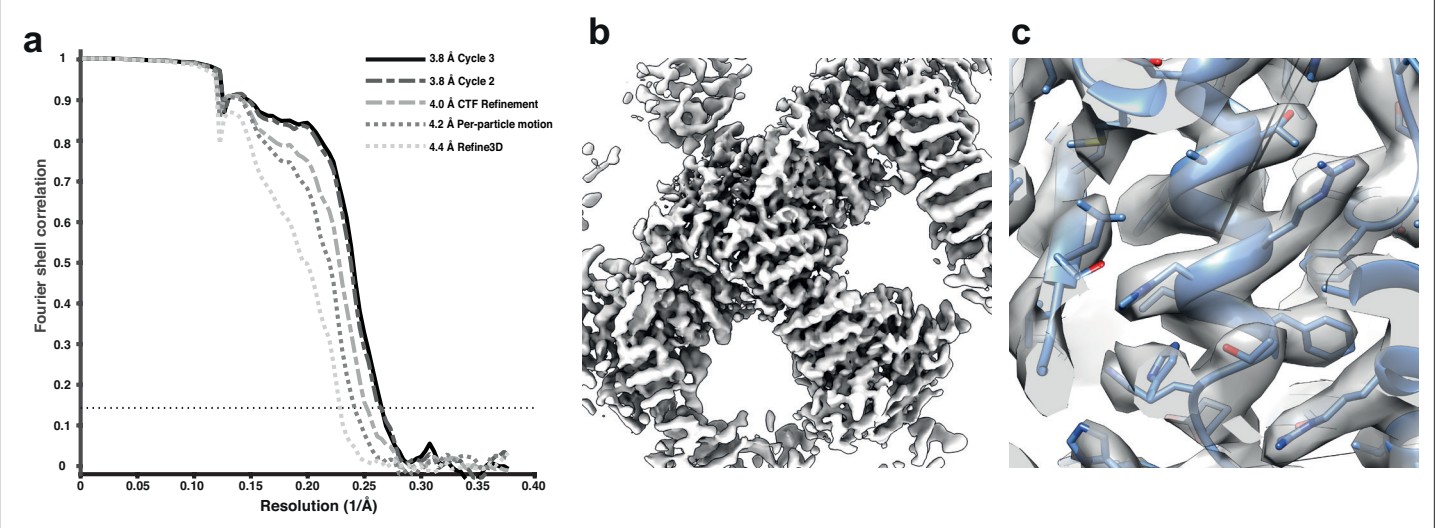

**Figure 3.** Subtomogram averaging of the COP-II inner layer. (**a**) FSC for resolution estimation of iteratively improved reconstructions using the new RELION-4.0 workflow, tested on the COP-II inner layer. (**b**) Reconstructed density for the inner layer. (**c**) Zoomed-in region of the final map (in transparent grey) with the refined atomic model (blue).

alignment and CTF refinement, respectively. Two further rounds of 3D refinement, followed by tilt-series frame alignment and CTF refinement, yielded a final map with a resolution of 3.8 Å (*Figure 3*).

At this resolution, most side chains are visible in the map, enabling us to build and refine an atomic model. The improved model will allow the design of point mutants to precisely disrupt interfaces between coat subunits and test their effects in COPII budding.

## Discussion

We formulate the problem of averaging over multiple identical particles in tomographic tilt series in an empirical Bayesian framework that is based on a statistical model that approximates one for two-dimensional experimental data. The Bayesian framework has proven effective in reducing the number of tunable parameters and in obtaining high-quality reconstructions from single-particle data (*Fernandez-Leiro and Scheres, 2016*). The two-dimensional data model describes the experimental images better than alternative approaches that use 3D reconstructed subtomograms as an inter-mediate. One example of a problem with the intermediate 3D data model is the need for missing wedge correction, which arises from the observation that the experimental images were acquired, incompletely, in three dimensions. Artefacts related to suboptimal missing wedge correction may affect both alignment and classification of particles. By using an approximation to the 2D data model, missing wedge correction is no longer required. Instead, the problem approaches that of single-particle analysis, where projections from different orientations and of different structural states are sorted out simultaneously. Provided the 3D Fourier transform of the distinct classes is fully sampled through the orientation distribution of the raw particles, likelihood optimisation techniques have been highly successful in tackling this problem in single-particle analysis (*Scheres et al., 2007*; *Fernandez-Leiro and Scheres, 2016*).

In practice, the implementation in RELION-4.0 does not use stacks of 2D projection images as input for the refinement program that performs alignment and classification. Instead, the concept of 3D pseudo-subtomograms is introduced, where the tilt-series images are Fourier transformed, pre-multiplied with their CTF, and inserted as a slice into a 3D Fourier volume according to the best current estimates for the tilt-series geometry. The use of 3D pseudo-subtomograms allowed reusing existing code for subtomogram averaging in RELION, while input stacks of 2D images would have required significant software development efforts. Nevertheless, in the future we might still choose to implement a true 2D version of the code, which would be more efficient, both in terms of processing time and disk storage requirements. In cases where the number of tilt images is small in compari-son to the box size, fewer Fourier pixels need to be examined in a stack of 2D images than in a

pseudo-subtomogram, with a corresponding decrease in processing time. Moreover, the likelihood calculation from the 3D pseudo-subtomogram approach requires separate storage of the accumulated squares of the CTFs, and the corresponding multiplicity terms. In contrast, in the 2D approach, only the 2D images need to be stored, as CTF parameters can be calculated on-the-fly and there is no need for a multiplicity term, giving a corresponding decrease in storage requirements. However, if one were to collect tilt series with very fine angular increments or in a continuous manner (*Chreifi et al., 2019*), then the current implementation may still be preferable.

Besides the alignment and classification of individual particles, the methods described here also deal with re-estimation of parameters that describe the optical and geometrical features of the tilt series. As soon as an initial average structure has been obtained, its increased signal-to-noise ratio can be exploited to determine these parameters more accurately than what is possible from the raw tilt-series images alone. The implementations in RELION-4.0 again follow those previously implemented for single-particle analysis, where CTF refinement is used for re-estimation of the tilt-series images CTFs, and algorithms akin to Bayesian polishing are used to re-estimate the tilt-series alignment, as well as the movement of individual particles throughout the tilt-series acquisition process. As better tilt-series parameters will allow better pseudo-subtomograms, particle alignment and classification are iterated with the optimisation of the tilt-series parameters.

Similar tilt-series and CTF optimisation approaches have been implemented in the program M (*Tegunov et al., 2021*). Compared to M, RELION-4.0 uses computationally more efficient algorithms; M uses GPUs to accelerate the calculations. In both tomography and SPA, RELION-4.0 only models beam-induced translations of the particles, whereas M also models beam-induced rotations. Since SPA routinely reaches 2 Å resolutions without modelling beam-induced rotations, we assumed that the effect of rotations of individual particles throughout the tilt series is not large enough to warrant their correction at typical tomography resolutions. In cases where the data do allow for better than 2 Å resolutions, M could still be used to correct for beam-induced rotations in a postprocessing step, following alignment and classification of the individual particles in RELION. It is likely that adaptation of M, in order to function with the pseudo-subtomograms proposed here, would lead to increased synergy between the two programs. In the meantime, external tools to convert from M parameters to RELION-4.0 are already available (https://github.com/joton/reliontomotools; *Zivanov, 2022* copy archived at swh:1:rev:bfa43828876ceb77bed0c7eb72f794c79c9de5e6).

Besides the reduction in tunable parameters that is characteristic of the Bayesian approach, its uptake by researchers that are new to the field is further facilitated through the implementation of a graphical user interface. This interface is already widely used for single-particle analysis and has been extended for the processing of tomographic data in RELION-4.0. Apart from the calculations that will be familiar to users of single-particle analysis, for example, 3D classification, 3D initial model generation, and 3D auto-refinement, the new interface also provides convenient access to the tomography-specific versions for CTF refinement and Bayesian polishing, as well as preprocessing operations to calculate the pseudo-subtomograms. However, tilt-series alignment, tomogram reconstruction, and particle picking are not yet part of the RELION workflow. Efforts to also implement solutions for those steps in a single tomography processing pipeline are ongoing and will be part of future RELION releases. Meanwhile, current import procedures rely on specific preprocessing operations in IMOD (*Kremer et al., 1996*), and particle coordinate conversion tools to use in RELION-4.0 are available for a range of third-party software packages (*Pyle et al., 2022*). To further facilitate the uptake of this new software by the community, we have provided an online tutorial that uses the publicly available HIV-1 immature capsid data set to describe and illustrate all steps necessary to obtain the results described in *Figure 1*.

In summary, we introduce new methods for subtomogram averaging to resolutions that are sufficient for de novo atomic modelling and increase the accessibility of this emerging technique. We envision that our methods will allow more researchers to calculate better structures from tomographic data, which will aid the next revolution in structural biology, where macromolecular complexes are imaged, not in isolation, but in their biologically relevant environment.

## Acknowledgements

We are grateful to the MRC-LMB EM facility for help with data acquisition and to Jake Grimmett, Toby Darling, and Ivan Clayson for help with high-performance computing. This work was funded

by the UK Research and Innovation (UKRI) Medical Research Council (MC_UP_A025_1013 to SHWS; and MC_UP_1201/16 to JAGB), the European Research Council (ERC) under the European Union's Horizon 2020 research and innovation program (ERC-CoG-2014, grant 648432, MEMBRANEFUSION to JAGB and ERC StG-2019, grant 852915 CRYTOCOP to GZ); the Swiss National Science Foundation (grant 205321_179041/1 to DC-D), the Max Planck Society (to JAGB) and the UKRI Biotechnology and Biological Sciences Research Council (grant BB/T002670/1 to GZ). TAMB is a recipient of a Sir Henry Dale Fellowship, jointly funded by the Wellcome Trust and the Royal Society (202231/Z/16/Z). JZ was partially funded by the European Union's Horizon 2020 research and innovation program (ERC-ADG-2015, grant 692726, GlobalBioIm to Michael Unser). TAMB thanks the Vallee Research Foundation, the Leverhulme Trust, and the Lister Institute of Preventative Medicine for support.

## Additional information

### Competing interests

Giulia Zanetti, Sjors HW Scheres: Reviewing editor, *eLife*. The other authors declare that no competing interests exist.

### Funding

| Funder | Grant reference number | Author |
|---|---|---|
| UK Research and Innovation | MC_UP_A025_1013 | Sjors HW Scheres |
| UK Research and Innovation | MC_UP_1201/16 | John AG Briggs |
| European Research Council | ERC-CoG-2014 grant 648432 | John AG Briggs |
| European Research Council | ERC-StG-2019 grant 852915 | Giulia Zanetti |
| Swiss National Science Foundation | 205321_179041/1 | Daniel Castaño-Díez |
| UK Research and Innovation | BBSRC grant BB/T002670/1 | Giulia Zanetti |
| European Research Council | ERC-AdG-2015 grant 692726 | Jasenko Zivanov |

The funders had no role in study design, data collection and interpretation, or the decision to submit the work for publication.

### Author contributions

Jasenko Zivanov, Joaquín Otón, Conceptualization, Software, Formal analysis, Investigation, Methodology, Writing – original draft, Writing – review and editing; Zunlong Ke, Formal analysis, Investigation, Visualization, Methodology, Writing – original draft, Writing – review and editing; Andriko von Kügelgen, Data curation, Formal analysis, Investigation, Visualization, Methodology, Writing – original draft, Writing – review and editing; Euan Pyle, Software, Formal analysis, Investigation, Visualization, Methodology, Writing – original draft, Writing – review and editing; Kun Qu, Formal analysis, Investigation, Methodology, Writing – review and editing; Dustin Morado, Software, Investigation, Methodology, Writing – review and editing; Daniel Castaño-Díez, Supervision, Funding acquisition, Project administration, Writing – review and editing; Giulia Zanetti, Tanmay AM Bharat, Formal analysis, Supervision, Funding acquisition, Investigation, Visualization, Writing – original draft, Project administration, Writing – review and editing; John AG Briggs, Conceptualization, Formal analysis, Supervision, Funding acquisition, Investigation, Visualization, Writing – original draft, Project administration, Writing – review and editing; Sjors HW Scheres, Conceptualization, Resources, Data curation, Software, Formal analysis, Supervision, Funding acquisition, Validation, Investigation, Methodology, Writing – original draft, Project administration, Writing – review and editing

## Author ORCIDs

Jasenko Zivanov (iD) http://orcid.org/0000-0001-8407-0759
Joaquín Otón (iD) http://orcid.org/0000-0002-2195-4730
Zunlong Ke (iD) http://orcid.org/0000-0002-8408-850X
Andriko von Kügelgen (iD) http://orcid.org/0000-0002-0017-2414
Euan Pyle (iD) http://orcid.org/0000-0002-4633-4917
Giulia Zanetti (iD) http://orcid.org/0000-0003-1905-0342
Tanmay AM Bharat (iD) http://orcid.org/0000-0002-0168-0277
John AG Briggs (iD) http://orcid.org/0000-0003-3990-6910
Sjors HW Scheres (iD) http://orcid.org/0000-0002-0462-6540

## Decision letter and Author response

Decision letter https://doi.org/10.7554/eLife.83724.sa1
Author response https://doi.org/10.7554/eLife.83724.sa2

## Additional files

### Supplementary files
• MDAR checklist

### Data availability

We have only used previously published cryo-EM data sets for testing our software. Reconstructed maps and atomic models generated in this study have been submitted to the EMDB and PDB, with entry codes as indicated in Table 1.

The following previously published dataset was used:

| Author(s) | Year | Dataset title | Dataset URL | Database and Identifier |
|---|---|---|---|---|
| Schur FK, Obr M, Hagen WJ, Wan W, Jakobi AJ, Kirkpatrick JM, Sachse C, Kräusslich HG, Briggs JA | 2018 | Cryo-electron tomography of immature HIV-1 dMACANC VLPs | https://www.ebi.ac.uk/empiar/EMPIAR-10164/ | Empiar, EMPIAR-10164 |

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

## Appendix 1

## CTF refinement

CTF refinement in RELION-4.0 optimises the following parameters: scale, defocus, astigmatism, and higher-order (even and odd) optical aberrations. Since, save for the difference in defocus, the same CTF needs to be valid for an entire micrograph of particles, similar optimisations can be applied as in our single-particle algorithms. All the relevant information is first consolidated into a minimal form using linear transformations, and the final, typically non-linear, optimisation is then performed on that minimal form.

We formulate the CTF for tilt-series frame $f$ of particle $p$ as follows:

$$\text{CTF}_{pf}(\mathbf{j}) = -\alpha_f \tau_f(\mathbf{j}) \sin(\gamma_{pf}(\mathbf{j})) \exp(i\rho_f(\mathbf{j})), \tag{14}$$

where $\alpha_f$ describes the overall scaling factor, $\tau_f(\mathbf{j})$ the empirical radiation-damage model as defined by *Grant and Grigorieff, 2015*, $\gamma_{pf\mathbf{j}}$ the symmetrical phase delay component, and $\rho_{f\mathbf{j}}$ the antisymmetrical one. Note that only $\gamma$ varies between particles. This is because it contains the quadratic defocus term that depends on the position of the particle. The phase delays are parametrised the same way as in single-particle analysis in RELION-3 – as a combination of explicitly named low-order terms and higher-order Zernike polynomials:

$$\gamma_{pf}(\mathbf{j}) = \pi\lambda\mathbf{j}^\mathsf{T} D_{pf}\mathbf{j} + \frac{\pi}{2}C_s\lambda^3|\mathbf{j}|^4 - \chi_f + \sum Z_n^m(\mathbf{j}), \tag{15}$$

$$D_{pf} = \begin{bmatrix} \delta z_p + a1 & a_2 \\ a_2 & \delta z_p - a2 \end{bmatrix} \tag{16}$$

As before, the astigmatic-defocus matrix $D_{pf}$ is decomposed into a defocus term $\delta z_p$ and two linear astigmatism terms, $a_1$ and $a_2$, while $C_s$ describes the spherical aberration of the microscope, $\chi_f$ a constant phase offset (owing to amplitude contrast and a phase plate, if one is used), $\lambda$ is the wavelength of the electron, and $Z_n^m$ are the higher-order even Zernike terms. One key difference to single-particle analysis is that the defocus term $\delta z_p$ is no longer a free parameter for each particle, but it instead depends on the already known 3D position of the particle. Therefore, in tomography, the defocus term is only estimated once per tilt image, and all the particles contribute to that estimate.

The scaling factor $\alpha_f$ is estimated by computing the following two sums for each micrograph and dividing them (the † symbol indicates complex conjugation):

$$\alpha_f = \frac{G_f}{H_f} \tag{17}$$

$$G_f = \sum_{p,\mathbf{j}} \frac{1}{\sigma_{|\mathbf{j}|}^2} \text{Re}(X_{pf\mathbf{j}}(\text{CTF}'_{pf}(\mathbf{j})V_{pf\mathbf{j}}^{(p)})^\dagger) \tag{18}$$

$$H_f = \sum_{p,\mathbf{j}} \frac{1}{\sigma_{|\mathbf{j}|}^2} |\text{CTF}'_{pf}(\mathbf{j})V_{pf\mathbf{j}}^{(p)}|^2. \tag{19}$$

Note that the CTF′ used in *Equation 19* is missing its scale factor:

$$\text{CTF}'_{pf}(\mathbf{j}) = -\tau_f(\mathbf{j}) \sin(\gamma_{pf}(\mathbf{j})) \exp(i\rho_f(\mathbf{j})). \tag{20}$$

Alternatively, we also allow the user to fit the parameters of Lambert's extinction model to the data instead, assuming perfectly flat samples of constant thickness. In that case, the CTF scale in image $f$ of tomogram $t$ is expressed as a function of the beam luminance $\alpha_0$, sample normal $\mathbf{n}_t$, and optical sample thickness $\kappa_t$:

$$\alpha_{tf}(\alpha_0, \kappa_t, \mathbf{n}_t) = \alpha_0 \kappa_t^{\frac{1}{|\mathbf{n}_t \cdot \mathbf{z}_f|}} \tag{21}$$

If this option is used, then the CTF scales of all the tilt series in the data set are estimated together. The beam luminance $\alpha_0$ is modelled globally, while the sample thickness and normal are allowed to differ among tomograms, but not between the images of a tilt series. The vector $\mathbf{z}_f$ points in viewing direction of tilt image $f$. Note that this model does not allow for separating the geometrical sample thickness from its extinction factor, so we can only estimate the product of the two. Also, the ice normal is required to be perpendicular to the estimated tilt axis of the tilt series since its component pointing in the direction of the axis is indistinguishable from an increase in ice thickness or opacity. This global optimisation is performed using the sums $G_{tf}$ and $H_{tf}$ computed in *Equations 18 and 19*, where the subscript $t$ indicates tilt-series $t$. This is done by finding a global value of $\alpha_0$ and values of $\kappa_t$ and $\mathbf{n}_t \in \mathbb{S}^2$ for all tomograms that produce $\alpha_{tf}$ which minimise the following quantity and thus maximise the overall likelihood in *Equation 4*:

$$\sum_{t,f} (H_{tf}\alpha_{tf}(\alpha_0, \kappa_t, \mathbf{n}_t) - G_{tf})^2.$$ (22)

To perform defocus estimation efficiently, we apply the optimisations we originally developed for the estimation of higher-order aberrations in single-particle analysis (*Zivanov et al., 2020*). It allows us to determine a collective offset to $\gamma$ for a large set of particles that all have different values of $\gamma$. Specifically, it allows the change to the log-likelihood arising from changing the value of $\gamma$ at any Fourier pixel to be expressed as a pair of 2D images, independently of the number of particles. Therefore, each pixel of each particle only needs to be considered once. After that, the log-likelihood can be evaluated by iterating over the pixels of a single image.

In single-particle analysis, this approach is used to estimate the higher-order aberrations that are shared among all the particles in a data set. In tomography, we also use this approach to condense the information from all the particles in a tilt image (all of which exhibit slightly different defoci), into two such images, and to then determine the optimal change to $\gamma$ efficiently using a nonlinear algorithm.

The two condensed images $R$ and $\hat{\mathbf{t}}$ that we compute are the same as the ones in single-particle analysis, except for the inclusion of the noise power $\sigma^2$. The definitions are repeated here for the sake of completeness. Note that each pixel $\mathbf{j}$ of $R$ contains a real symmetrical $2 \times 2$ matrix and each pixel of $\hat{\mathbf{t}}$ a $\mathbb{C}^2$ vector:

$$R_{f\mathbf{j}} = \sum_p \frac{1}{\sigma_j^2} |\widetilde{V}_{f\mathbf{j}}^{(p)}|^2 \mathbf{d}_{pf\mathbf{j}} \mathbf{d}_{pf\mathbf{j}}^{\mathsf{T}}$$ (23)

$$\hat{\mathbf{t}}_{f\mathbf{j}} = -R_{f\mathbf{j}}^{-1} \sum_p \frac{1}{\sigma_j^2} \mathrm{Re}(X_{pf\mathbf{j}}^{\dagger} \widetilde{V}_{f\mathbf{j}}^{(p)}) \mathbf{d}_{pf\mathbf{j}},$$ (24)

where $\mathbf{d}_{pf\mathbf{j}} \in \mathbb{R}^2$ describes the point on the unit circle corresponding to the initial phase angle $\gamma^{(0)}$, which is given by the initial CTF parameters:

$$\mathbf{d}_{pf\mathbf{j}} = \begin{bmatrix} \cos(\gamma_{pf}^{(0)}(\mathbf{j})) \\ \sin(\gamma_{pf}^{(0)}(\mathbf{j})) \end{bmatrix}$$ (25)

The predicted 2D images $\widetilde{V}_f^{(p)}$ contain the effect of the initial CTFs, except for the symmetrical aberration:

$$\widetilde{V}_{f\mathbf{j}}^{(p)} = -\alpha_f \tau_f(\mathbf{j}) \exp(i\rho_f(\mathbf{j})) V_{f\mathbf{j}}^{(p)}$$ (26)

The vector-valued condensed image $\hat{\mathbf{t}}_f$ describes the most likely phase shift $\gamma$ for each pixel $\mathbf{j}$, expressed as a point on a circle, while the matrix-valued one, $R_f$, describes the anisotropic weight of that information. With these two condensed images computed for a given tilt image $f$, the change to the likelihood defined in *Equation 4* resulting from the change to the phase delay $\gamma$ at any pixel can be expressed as a quadratic form. Therefore, we look for a change $\delta D$ to the astigmatic-defocus matrices $D_{pf}$ which produces phase delays that minimise that quadratic form:

$$C_f = \sum_{\mathbf{j}} \mathbf{e_j^T}(\delta D) R_{\mathbf{j}} \mathbf{e_j}(\delta D), \tag{27}$$

where the per-pixel error $\mathbf{e_j}(\delta D)$ is given by the deviation from the optimal phase shift $\widehat{\mathbf{t_j}}$:

$$\mathbf{e_j}(\delta D) = \begin{bmatrix} \cos(\delta\gamma) - \mathrm{Re}(\widehat{\mathbf{t_j}}) \\ \sin(\delta\gamma) - \mathrm{Im}(\widehat{\mathbf{t_j}}) \end{bmatrix} \tag{28}$$

$$\delta\gamma = \mathbf{j^T} \delta D \mathbf{j} \tag{29}$$

As an alternative to fitting $D$ independently for each tilt-series image, our program also allows the user to apply an $L^2$ regulariser to the $\delta D_f$ of the different images in the same series. In that case, the sum in *Equation 27* runs over all the pixels $\mathbf{j}$ of all the tilt-series frames $f$. This helps to stabilise the CTFs of the higher tilts, but it risks impairing the estimates of the CTFs of the more important lower tilts. Formally, this is done by minimising the following cost:

$$C_{\mathrm{glob}} = \sum_{f,\mathbf{j}} \mathbf{e_{f\mathbf{j}}^T}(D_f) R_{f\mathbf{j}} \mathbf{e_{f\mathbf{j}}}(D_f) + \lambda \sum_f |D_f - \widehat{D}|^2, \tag{30}$$

Since the early tilt-series images carry more information than the later ones, their values in $R$ are typically significantly greater. Therefore, using this formulation, they automatically assume a greater weight in the estimation. The optimal weight for the regulariser itself, $\lambda$, cannot be measured from the data. Its optimal value depends on how reproducible the defocus values are for each of the tilt-series images, which in turn depends on the microscope setup, such as the stability of the stage.

## Geometric refinement

Analogously to Bayesian polishing, the log-likelihood of a particle being observed at a position $\mathbf{s}$ is given by twice its cross-correlation with the predicted image:

$$\log(P(X|\mathbf{s})) = 2\mathrm{CC}(\mathbf{s}) \tag{31}$$

$$CC = IFT(\omega X V^{(P)\dagger}) \tag{32}$$

$$w_{\mathbf{j}} = \frac{1}{\sigma_j^2}, \tag{33}$$

where IFT stands for inverse Fourier transform.

To keep the problem differentiable, the cross-correlation CC is always accessed using cubic interpolation. After the inverse Fourier transformation, each such cross-correlation table is cropped to a smaller size to make storing many such tables feasible, and the memory throughput efficient. The size of the tables can be controlled by the user, and should be set to the maximal positional error expected in the data set.

The geometrical model that is optimised this way projects points $\mathbf{s} \in \mathbb{R}^3$ in the tomogram to 2D positions $\mathbf{p}_f \in \mathbb{R}^2$ in each tilt image:

$$\mathbf{p}_f = W(\mathbf{1}_f) \tag{34}$$

$$I_f = J_f \begin{bmatrix} s \\ 1 \end{bmatrix} \tag{35}$$

The initial linear projection $\mathbf{l}_f$ is obtained by multiplying $\mathbf{s}$ with a $2 \times 4$ matrix $J_f$, and then optionally shifted by the non-linear image distortion $W$. The cost $C_{\mathrm{align}}$ that is being minimised consists of the sum over all (negative) cross-correlation values of all particles in all images plus all regularisers for all regularised parameters:

$$C_{\mathrm{align}} = -\sum_{p,f} \mathrm{CC}(\mathbf{p}_f - \mathbf{p}_f^{(0)}) + \mathfrak{R} \tag{36}$$

Although our framework supports arbitrary projection matrices $J_f$, our optimisation algorithm only looks for orthogonal rotations to the initial projection matrix. This is achieved by parametrising that rotation using Tait–Bryan angles, not Euler angles. The disadvantage of Euler angles is that they are gimbal locked in the initial configuration where all three angles are zero, that is, the first and third angles refer to the same axis. The rigid alignment of the tilt image is never regularised because we do not assume to have any prior information on it.

The distortion field $W$ can take different forms. We have implemented models that express the distortion using the Fourier basis, a cubic spline basis, and an affine linear one. The intended purpose of these deformations is to model distortions of the image that arise at the image forming lenses at the bottom of the optical system. An imperfect calibration of these lenses is likely to go unnoticed as long as the microscope is only used for single-particle analysis because the same particle is never observed at starkly different positions during the collection of a single-particle movie. In tomography, a given particle may appear at any image position in any tilt image, so arbitrary deformations to the 2D image become relevant. We expect these deformations to be stable over time, so the intended purpose of the deformation field $W$ is to model only one such deformation per tilt series. Optionally, we also allow the user to instead model a different deformation for each tilt image, but we have not encountered any data sets where this has produced an improvement. The deformation fields are optionally regularised by penalising the squared coefficients of the respective model. This limits the extent of deformation, and it forces the system to explain changes in position through particle motion, rather than image deformations.

The quantity that we do expect to change during the collection of the tilt series is instead the 3D position of the particle, $\mathbf{s}_f$. Analogously to Bayesian polishing, we model this change as motion over time. The position in image $f$ is given by the sum over its per-tilt-series-frame velocities $\mathbf{v}_f \in \mathbb{R}^3$ up to that point. Note that the velocity vector $\mathbf{v}_f$ refers to the motion between images $f$ and $f+1$:

$$S_f = S_0 + \sum_{f'=0}^{f' < f} v_f \tag{37}$$

It is important to note that the tilt images are implicitly assumed to be ordered chronologically. In practice, this is usually not given, so the images are reordered by the program based on the cumulative radiation dose of each image.

As in Bayesian polishing, the motion vectors themselves are expressed in a collective coordinate system for all the particles. This allows the spatial smoothness of a hypothetical motion field to be evaluated and used as a prior. For a more detailed derivation, we refer to the paper on Bayesian polishing for single-particle analysis (*Zivanov et al., 2019*). The formal details will be given in the following for the sake of completeness and to highlight differences to the original formulation.

The collective coordinate system for particle motion is obtained through a low-rank approximation of a Gaussian process. This is done by constructing and then diagonalising the $P \times P$ covariance matrix $S$ for a set of initial particle positions (where $P$ is the number of particles). The entries of $S$ contain the value of the following square-exponential covariance function for each pair of particles $p$ and $q$:

$$S_{p,q} = \sigma_V^2 \exp(-|\mathbf{s}_p - \mathbf{s}_q|^2/\sigma_D). \tag{38}$$

Optionally, the user can instead also use the original formulation without the square inside the exponential:

$$S'_{p,q} = \sigma_V^2 \exp(-|\mathbf{s}_p - \mathbf{s}_q|/\sigma_D). \tag{39}$$

The former option forces particles in immediate proximity to move more similarly, but it allows for a greater discrepancy at greater distances. Both the single-particle and the tomography implementations allow the user to choose either function, but the default has changed from the latter to the former in tomography. This choice was motivated by both empirical observations and the fact that the square-exponential kernel produces fewer meaningful deformation components, which speeds up the optimisation for tomograms with hundreds of particles.

We perform a singular-value decomposition of the covariance matrix $S$,

$$S = U\Lambda W^{\mathsf{T}}, \tag{40}$$

which allows us to construct the coordinate system as follows:

$$\mathbf{b}_i = \sqrt{\lambda_i}\mathbf{w}_i, \tag{41}$$

where $\lambda_i \in \mathbb{R}$ is the $i^{th}$ singular value and $\mathbf{w}_i \in \mathbb{R}^P$ the corresponding singular vector. Basis vectors with small $\lambda_i$ are discarded here to speed up the optimisation. Let $P'$ represent the number of remaining basis vectors. In this coordinate system, the set of all particle velocities in a tilt image $V_f \in (\mathbb{R}^3)^P$ can be expressed as $V_f = BQ_f$, where $Q_f$ is a $P' \times 3$ coefficient matrix that encodes the velocity in three spatial dimensions. Note that the same basis $B$ is shared between all three dimensions and all tilt-series frames. In this coordinate system, the negative log-likelihood of a configuration of particle velocities is given by the Frobenius norm of the coefficient vector, $\|Q_f\|$, that is, the sum of the squares of its entries. Therefore, the motion regulariser takes a simple form:

$$\mathfrak{R}_{\mathrm{motion}} = \sum_f \|Q_f\|. \tag{42}$$

The acceleration regulariser that would penalise changes in velocity from one tilt-series frame to the next in single-particle analysis has been omitted from tomography. This is because, unlike a single-particle movie, the tilt images are not collected in one continuous exposure. Since they are being exposed individually, there is no reason to assume that the particle velocities will be continuous between them. Two further differences to the original Bayesian polishing are hidden in the notation: the covariance is now based on the 3D distances between the particles, and the coefficient matrix $Q$ contains three columns instead of two.

As in the original Bayesian polishing approach, the complete alignment of the tilt series is performed by finding parameters that minimise $C_{\mathrm{align}}$ from *Equation 36* using L-BFGS (*Liu and Nocedal, 1989*). The set of parameters that are being optimised always includes the three Tait–Bryan angles for each tilt image and the set of initial particle positions. The latter are essential because all the information we have about their 3D positions is derived from the tilt images themselves, so changing the alignment requires the particles to be able to shift to more likely positions. Estimating the image deformations and particle motion is optional. If they are being estimated, then a set of deformation coefficients is fitted either to each tilt image or to each tilt series, while a set of motion coefficients is fitted to each image transition.

In addition to this local, L-BFGS-based refinement, we also offer two methods to align only the 2D shifts of all tilt images globally. This means that instead of trying to obtain the optimal alignment through small changes to the initial one, we instead look for the best possible image shift overall, keeping all other parameters constant. This is helpful when individual tilt images are so badly aligned that a local optimisation cannot converge to the globally optimal position. Note that the initially assumed angles are rarely as incorrect as the image shifts since the angles can be controlled more effectively through the experimental setup.

There are two variants to this method. If the sample contains few particles per tomogram, then the best results are obtained by predicting an entire micrograph and computing its cross-correlation with the original one. The maximum value in that large cross-correlation image then indicates the optimal image shift. This approach can in theory deal with arbitrarily large misalignments. If the sample is dense, however, then this whole-micrograph approach can fail. In that case, better results are obtained by instead adding up the small, per-particle cross-correlation images defined in *Equation 31*, and finding the maximum in that sum. This latter approach can only correct for misalignments smaller than half the box size of the particle, and it often produces inferior results on samples with few particles per tomogram.

## Appendix 2

**Appendix 2—table 1.** Electron cryo-tomography (Cryo-ET) data collection, refinement, and validation statistics.

| | HIV-1 Gag (EMD-16207 /EMD-16209) | S-layer_inner_domain (EMD-16183) (PDB 8BQE) | COPII inner coat (EMD-15949) (PDB 8BSH) |
|---|---|---|---|
| **Data collection** | | | |
| Microscope | Titan Krios | Titan Krios | Titan Krios |
| Detector | K2 (Gatan) | K2 (Gatan) | K2 (Gatan) |
| Software | SerialEM (*Mastronarde, 2003*) | SerialEM (*Mastronarde, 2003*) | SerialEM (*Mastronarde, 2003*) |
| Voltage (kV) | 300 | 300 | 300 |
| Slit width (eV) | 20 | 20 | 20 |
| Defocus range (μm) | –1.5 to –5.0 | –1.5 to –5.0 | –1.5 to –4.5 |
| Pixel size (Å) | 1.35 | 1.35 | 1.33 |
| Total exposure ($e^-$/Å$^2$) | 120–145 | ~140 | ~120 |
| Exposure per tilt ($e^-$/Å$^2$) | 3.0–3.5 | 3.4 | 2.9 |
| Total number of tilts | 41 | 41 | 41 |
| Frames per tilt-movie | 8–10 | 10 | 10 |
| Tilt increment | $\pm3°$ | $\pm3°$ | $\pm3°$ |
| Tilt-series scheme | Dose-symmetrical | Dose-symmetrical | Dose-symmetrical |
| Tilt range | $\pm60°$ | $\pm60°$ | $\pm60°$ |
| Tilt-series (no.) | 5/43 | 110 | 137 |
| **Data processing** | | | |
| Software tilt-series alignment | IMOD (*Kremer et al., 1996*) | IMOD (*Kremer et al., 1996*) | Dynamo (*Castaño-Díez et al., 2012*) |
| Software CTF estimation | CTFPLOTTER (*Xiong et al., 2009*) | CTFFIND4 (*Rohou and Grigorieff, 2015*) | CTFFIND4 (*Rohou and Grigorieff, 2015*) |
| Particle images (no.) | 12,910/144,275 | 42,990 | 106,533 |
| Pre-cropped box-size (pix) | 512 | 600 | 512 |
| Final box-size (pix) | 192 | 180 | 196 |
| Pixel size final rec. (Å) | 1.35 | 1.35 | 1.33 |
| Symmetry imposed | C6 | C6 | C1 |
| Map resolution (Å) | 3.2/3.0 | 3.5 | 3.8 |
| FSC threshold | 0.143 | 0.143 | 0.143 |
| Map resolution range (Å) | 3.2–4.3/3.0–3.5 | 3.5–4.8 | 3.8–7.2 |
| Map sharpening B factor (Å$^2$) | –85 / –95 | –75 | –106 |
| **Model refinement** | | | |
| Initial model used (PDB code) | | 6T72 | 6GNI |

*Appendix 2—table 1 Continued on next page*

*Appendix 2—table 1 Continued*

| | HIV-1 Gag (EMD-16207 /EMD-16209) | S-layer_inner_domain (EMD-16183) (PDB 8BQE) | COPII inner coat (EMD-15949) (PDB 8BSH) |
|---|---|---|---|
| Software | | PHENIX (*Afonine et al., 2018*) | Isolde (*Croll, 2018*) and PHENIX (*Afonine et al., 2018*) |
| Model resolution (Å) | | 3.6 | 4.0 |
| FSC threshold | | 0.5 | 0.5 |
| **Model composition** | | | |
| Non-hydrogen atoms (no.) | | 11,274 | 13,635 |
| Protein residues (no.) | | 1452 | 1729 |
| **R.m.s. deviations** | | | |
| Bond lengths (Å) | | 0.001 | 0.004 |
| Bond angles (°) | | 0.322 | 0.858 |
| **Validation** | | | |
| MolProbity score | | 1.13 | 2.01 |
| Clashscore | | 3.42 | 14.71 |
| Poor rotamers (%) | | 0 | 0.66 |
| $C\beta$ outliers (%) | | 0 | 0.00 |
| CABLAM outliers (%) | | 0.84 | 2.33 |
| **Ramachandran plot** | | | |
| Favoured (%) | | 98.8 | 95.1 |
| Allowed (%) | | 1.2 | 4.8 |
| Disallowed (%) | | 0 | 0.1 |

**Appendix 2—table 2.** Computational costs and hardware specifics.

| | HIV-1 Gag | S-layer inner domain | COPII inner coat |
|---|---|---|---|
| Tilt-series (no.) | 5/43 | 110 | 137 |
| Final particle images (no.) | 12,910/144,275 | 42,990 | 106,533 |
| Pre-cropped box-size (pix) | 512 | 600 | 512 |
| Final box-size (pix) | 192 | 180 | 196 |
| **Computational costs** | | | |
| Pseudo-subtomogram | | | |
| Compute time | 21 min/40 min | 34 min | 67 min |
| Number of CPU nodes | 1/1 | 1 | 1 |
| Disk space | 343 GB/3.8 TB | 777 GB | 3.1 TB |
| Refine3D | | | |
| Compute time | 18 hr*/33 hr | 14 hr | 57 hr |
| Number of GPU nodes | 1/1 | 1 | 1 |
| Ctf refinement | | | |
| Compute time | 15 min*/35 min | 2 hr | 2 hr |
| Number of CPU nodes | 1/1 | 1 | 1 |

*Appendix 2—table 2 Continued on next page*

*Appendix 2—table 2 Continued*

|  | HIV-1 Gag | S-layer inner domain | COPII inner coat |
|---|---|---|---|
| Disk space | 32 MB/247 MB | 621 MB | 673 MB |
| Frame alignment |  |  |  |
| Compute time | 2 hr*/12 hr | 2 hr | 6 hr |
| Number of CPU nodes | 1/1 | 1 | 1 |
| Disk space | 383 MB/4.1 GB | 1.9 GB | 2.2 GB |
| **Hardware specifics** |  |  |  |
| CPU nodes |  |  |  |
| CPU model | 2x Intel Xeon E5-2698 v4 | 2x Intel Xeon 6258R | 1x AMD EPYC 7H12 |
| CPU memory | 512 GB | 754 GB | 256 GB |
| GPU nodes |  |  |  |
| CPU model | 2x Intel Xeon Silver 4116 | 2x Intel Xeon E5-2667 v4 | 1x AMD EPYC 7H12 |
| CPU memory | 384 GB | 256 GB | 256 GB |
| GPU model | 2x Nvidia Quadro RTX 5000 | 4x Nvidia GeForce GTX 1080 Ti | 4x Nvidia RTX A6000 |

*These calculations were performed using the same hardware as for the S-layer inner domain.

