## [Editor Report]

Single-particle tomography (SPT) is a useful method to determine the structure of proteins imaged in situ. This important work presents an easy-to-use tool for SPT that approximates the use of 2D tomographic projections using a ‘pseudo-subtomogram’ data structure, chosen to facilitate implementation within the existing RELION codebase. The examples shown provide solid support for the claims about the efficacy of the approach.

---

## [Decision Letter]

**Decision letter after peer review:**

Thank you for submitting your article "A Bayesian approach to single-particle electron cryo-tomography in RELION-4.0" for consideration by *eLife*. Your article has been reviewed by 3 peer reviewers, one of whom is a member of Our Board of Reviewing Editors, and the evaluation has been overseen by Volker Dötsch as the Senior Editor. The following individual involved in the review of your submission has agreed to reveal their identity: Alberto Bartesaghi (Reviewer #1).

Essential revisions:

(1) Half the citations in the manuscript are self-references (16 out of 32!), and important/relevant work on SPT was left out and should be referenced: Himes et al., 2018 (https://doi.org/10.1038/s41592-018-0167-z), Chen et al., 2019 (https://doi.org/10.1038/s41592-019-0591-8), and Sanchez et al., 2019 (https://doi.org/10.23919/EUSIPCO.2019.8903041). Also, with the exception of their own sub-volume averaging approach [4], the vast body of work on missing-wedge compensation was not referenced in the introduction.

(2) In the section on "Orientation priors", they state that one advantage of the proposed approach is that "the coordinate system of the pseudo subtomograms themselves can be chosen arbitrarily" and that "this not only accelerates the refinement" but also "makes it possible to solve more challenging structures because fewer solutions are allowed". This advantage, however, is certainly not specific to the "pseudo-subtomogram" data representation proposed here. Indeed, this strategy was originally proposed in the context of single-particle cryo-EM by Jinag et al., 2001 (https://doi.org/10.1006/jsbi.2001.4376) and was first used for processing cryo-ET data by Forster et al., 2005 (https://doi.org/10.1073/pnas.040917810), followed by many others. This point should be contextualized and the appropriate references included.

(3) Results on the HIV-1 Gag benchmark dataset (EMPIAR-10164) have two components: (1) processing of a "standard" subset consisting of 5 tilt-series, and (2) processing of the entire dataset (43 tilt-series). As far as I know, the best reconstruction obtained using the five tilt-series is a 3.3 A map obtained using emClarity (https://www.ebi.ac.uk/emdb/EMD-13354). Since only lower resolution results obtained with NovaCTF (3.9A) and Warp (3.8A) are cited in the text, my guess is that the authors were not aware of this newer result. Given that the resolution of the emClarity map (3.3A) is similar to the one presented here (3.2A), it would be useful to compare their reconstruction against emClarity's EMD-13354 (both in terms of map features and FSC).

(4) When processing the entire dataset, a 3.0A resolution map was obtained which matches the resolution obtained by Warp/M [24]. This result, however, is "not shown" (page 5). Since this is the only benchmark dataset analyzed in this study and given that many other packages have analyzed the same data, this map should be presented together with the corresponding FSC curve, data processing details, and comparisons made against M's 3.0A EMD-11655 map.

(5) No experiments are presented to validate the ability of the approach to correct for higher-order aberrations. Since "higher-order aberration correction will probably be most useful for reconstructions that extend beyond 3A resolution", why didn't they validate their approach using one of the sub-3A cryo-ET datasets available on EMPIAR (e.g., EMPIAR-10491)?

(6) It would be useful to include a table with a summary of all the relevant data collection/data processing parameters (e.g., pixel size, detector, number of tilt-series and sub-volumes, symmetry, resolution, number of refinement rounds, etc.) for all datasets analyzed in this study.

(7) Ony the 3.8A map for the COPII inner coat has been deposited in the EMDB. All other maps and related metadata need to be deposited as well.

(8) Computational complexity is a very important topic in the context of SPT, but no details are included in the text other than the following statement: "Compared to M, RELION-4.0 uses computationally more efficient algorithms that do not require the computational power of a GPU". Does this imply that computationally inefficient algorithms require the use of GPUs? If the authors want to comment on computational complexity (which I think would be very important to do), actual numbers need to be presented, such as running times, compute resources used, storage, etc.

9) Regarding the use of a "true 2D approach" vs. "pseudo-volumes" (page 6), the following claim is made: "For example, the current implementation could be used to process pairs of tilted images, but a 2D implementation would be more efficient", does this refer to efficiency in computing terms, storage, or both? Also, the authors should comment on the storage requirements of their 3D approach. Since the pseudo-tomograms have three components (Eqs. 8-10), does this mean that they require 3x more storage compared to traditional subtomogram averaging? This would be an important practical consideration for prospective users of this tool and should be adequately discussed.

10) In general, the use of the term "particle polishing" could lead to misinterpretations since it means different things in SPA and tomography. In SPA, "polishing" refers to the analysis of intermediate vidoe frames (which also exist in tomography), but the present study doesn't consider them at all. Also, I noticed that the term "frames" is used interchangeably to refer to SPA "vidoe frames" and "particle tilts" in tomography, which adds to the confusion. For example, the term "frame alignment" is mentioned twice on page 6, but I believe the authors are really referring to "tilt image alignment" instead. This should be clarified and the terms "frame" and "tilts" should be used consistently throughout the manuscript.

(11) For the *Caulobacter crescentus* S-layer reconstruction, the new cryo-ET map corresponding to the inner domain of the S-layer seems to have even better resolution than the SPA reconstruction reported in the original work [26] (3.5A vs. 3.7A for EMD-10389). Since the authors say these maps are in "excellent agreement", a quantitative comparison should be included. Was the second LPS binding site visible in the 3.7A SPA map? In Figure 2d, the fit between LPS and the cryo-ET density (right-hand side) seems to be of lower quality (i.e., no density for parts of the LPS model is seen). Without having access to the maps, it is not possible to properly evaluate these results.

(12) The paper is well-written and clear, but the description of the new concepts and evaluation of the method is somewhat minimalistic. While these may not be absolutely necessary, they will facilitate better understanding by a broader user community.

(i) A graph describing the workflow as implemented in RELION, e.g., as a flow chart, would increase the readability of the paper.

(ii) A figure visually describing the pseudosubtomograms would make the concept clearer to the reader.

(iii) A more in-depth description of the input files (alignment files from imod, particle lists, and orientation from Dynamo/elsewhere) needed to perform alignments and averaging in RELION using pseudosubtomograms would be beneficial to the community. This could be incorporated into a flowchart as suggested in (i).

13) Along the same lines, the authors do not include any description of the particle-picking procedures or the software requirements. This seems to be important as the authors state "(particles) relative defoci are known from the geometry of the tilt series and the known 3D positions of the particles in the tomogram".

14) In section 2.2 it is shortly discussed that rotational priors can be used to constrain orientational searches. As this is often very useful for particle alignment in cryo-ET, this and the implementation should be described in more detail.

15) The progress in map quality is only shown by improvements to resolution in the FSC for the test datasets. It would be beneficial to also show the initial and intermediary maps for at least one of the test cases so that the reader can better assess the improvement in map quality. Overall maps should be shown to evaluate performance on the "whole particle" level, given considerations of flexibility (especially in cases where focused refinement was used). Corresponding local resolution maps would be informative in such cases.

(16) For the immature HIV capsid data, the FSC curves are only shown for the small subset of data. As it is an important point that the RELION pseudosubtomogram approach can reach the same resolution as previous software such as M, the data should also be shown for the full dataset (authors write "data not shown").

(17) In general, a proper comparison to the previous approaches/results, maybe by providing the maps for comparison in the figures, would be helpful for the reader to evaluate which approach to choose, and based on what expectations/biological system.

18) In section 4 it is briefly discussed that particle rotation is not modeled in the per-particle motion correction. They draw parallels to single particles and conclude that this might be an issue at resolutions better than 2 Å. However, could the authors comment on whether the difference in dose and time frame of data collection as well as the thickness of sample, may affect the assumption that this is only an issue at a higher resolution?

(19) Many subtomogram averages are resolved at a more moderate resolution (e.g. 8-20 Å) than the maps presented in this study due to different reasons. It would be beneficial for the reader if the authors could comment on/discuss which optical and geometrical parameters can be confidently modeled at a lower resolution and if they improve resolution.

(20) Could the authors comment on whether it is possible to extract the refined optical and geometrical parameters for a set of particles from a tilt series, and apply them to a different set of particles from the same tilt series (or to the full tilt series to obtain a better quality tomogram for new particle picking).

(21) Along the same line: would it be possible to refine two distinct species (e.g. two distinct conformations of the same protein or two completely unrelated proteins) together in a multi-species refinement of optical and geometrical parameters? As the overall tilt series alignment and imaging parameters are the same, this might aid in refinement for lower abundance.

(22) On a more conceptual level, the reasoning to separate the SPA/Cryo-ET workflows completely in the software package remains unclear to me and it would be valuable if the authors could briefly discuss this.

(23) Can the authors comment on the "The data model assumes independent Gaussian noise on the Fourier components of the cryo-EM images of individual particles" with respect to the more complex in situ data, where noise is expected to be more structured?

---

## [Author Response]

Essential revisions:(1) Half the citations in the manuscript are self-references (16 out of 32!), and important/relevant work on SPT was left out and should be referenced: Himes et al., 2018 (https://doi.org/10.1038/s41592-018-0167-z), Chen et al., 2019 (https://doi.org/10.1038/s41592-019-0591-8), and Sanchez et al., 2019 (https://doi.org/10.23919/EUSIPCO.2019.8903041). Also, with the exception of their own sub-volume averaging approach [4], the vast body of work on missing-wedge compensation was not referenced in the introduction.

We have included the references above, and multiple references to alternative approaches for subtomogram averaging and approaches for missing-wedge correction. The revised manuscript now contains 53 references.

(2) In the section on "Orientation priors", they state that one advantage of the proposed approach is that "the coordinate system of the pseudo subtomograms themselves can be chosen arbitrarily" and that "this not only accelerates the refinement" but also "makes it possible to solve more challenging structures because fewer solutions are allowed". This advantage, however, is certainly not specific to the "pseudo-subtomogram" data representation proposed here. Indeed, this strategy was originally proposed in the context of single-particle cryo-EM by Jinag et al., 2001 (https://doi.org/10.1006/jsbi.2001.4376) and was first used for processing cryo-ET data by Forster et al., 2005 (https://doi.org/10.1073/pnas.040917810), followed by many others. This point should be contextualized and the appropriate references included.

This is a misunderstanding. We are not claiming that performing local orientational searches is new in our approach. Only the option to arbitrarily pre-orient the pseudo-subtomograms is new. This makes expressing local angular searches in terms of the Euler angles more straightforward. We have completely rewritten this section to reflect this more clearly; it now reads:

“Pre-oriented priors

Many proteins are organised in imperfect 2D arrays inside the tomograms, for example inside membranes or as part of capsid-like structures. Often, the individual protein molecules inside these arrays exhibit limited rotational freedom with respect to the surface normal of the array, although they may be able to rotate freely around that normal. This knowledge is often exploited in subtomogram averaging approaches through local orientational searches, e.g. see Forster2005retro. This not only accelerates the refinement, as fewer orientations need to be evaluated, it also makes it possible to solve more challenging structures because fewer solutions are allowed. In RELION, local orientational searches are implemented as Gaussian priors on the Cartesian translations and on the three Euler angles that describe rotations Scheres2012relion. One advantage of using pseudo subtomogram alignment is that the coordinate system of the pseudo subtomograms themselves can be chosen arbitrarily. By default, pseudo subtomograms are created in the same orientation as the tomogram, but the user can choose to orient them in a more meaningful way. For example, by constructing the pseudo subtomograms with their Z-axis parallel to the 2D array, using a rotational prior of approximately 90^o^ on the tilt angle will limit the amount of rocking of the particles inside the array, while avoiding singularities in the definition of the Euler angles that occur when the tilt angle is close to 0^o^.”

(3) Results on the HIV-1 Gag benchmark dataset (EMPIAR-10164) have two components:(1) processing of a "standard" subset consisting of 5 tilt-series, and (2) processing of the entire dataset (43 tilt-series). As far as I know, the best reconstruction obtained using the five tilt-series is a 3.3 A map obtained using emClarity (https://www.ebi.ac.uk/emdb/EMD-13354). Since only lower resolution results obtained with NovaCTF (3.9A) and Warp (3.8A) are cited in the text, my guess is that the authors were not aware of this newer result. Given that the resolution of the emClarity map (3.3A) is similar to the one presented here (3.2A), it would be useful to compare their reconstruction against emClarity's EMD-13354 (both in terms of map features and FSC).

We have included a reference to the new emClarity map in the Results section on the 5tomogram HIV data set. A comparison between our map and the emClarity map is now shown in Figure 1 —figure supplement 2. The RELION map is better than the emClarity map, which used a different, more optimistic mask for its original resolution estimation.

(4) When processing the entire dataset, a 3.0A resolution map was obtained which matches the resolution obtained by Warp/M [24]. This result, however, is "not shown" (page 5). Since this is the only benchmark dataset analyzed in this study and given that many other packages have analyzed the same data, this map should be presented together with the corresponding FSC curve, data processing details, and comparisons made against M's 3.0A EMD-11655 map.

We now show these results and a comparison with the map from M in Figure 1 —figure supplement 3. The two maps are very similar.

(5) No experiments are presented to validate the ability of the approach to correct for higher-order aberrations. Since "higher-order aberration correction will probably be most useful for reconstructions that extend beyond 3A resolution", why didn't they validate their approach using one of the sub-3A cryo-ET datasets available on EMPIAR (e.g., EMPIAR-10491)?

Higher-order aberration correction has now been used on a sub-2A resolution apo-ferritin test data set from a different collaborator. “Unfortunately”, their scope was so well-aligned that not even at this resolution aberration correction led to an improvement. Therefore, and because these collaborators are planning on writing their own publication about these results, we decided not to include these data in our revision.

We merely describe aberration correction for completeness and to make our potential readers aware of the available tools, as we do not intend to write a separate paper on this functionality. We also note that the application of fast, multi-shot data acquisition techniques for tomography that use beam shifts (like FISE and BISECT) may lead to wider applicability in the future, or least for those who do not align their microscope optimally…

(6) It would be useful to include a table with a summary of all the relevant data collection/data processing parameters (e.g., pixel size, detector, number of tilt-series and sub-volumes, symmetry, resolution, number of refinement rounds, etc.) for all datasets analyzed in this study.

We now include data acquisition details for each data set in Table 1.

(7) Only the 3.8A map for the COPII inner coat has been deposited in the EMDB. All other maps and related metadata need to be deposited as well.

The 5-tomo (EMD-16207) and 43-tomo (EMD-16209) maps of the HIV capsid lattice and the map and model for the *Caulobacter crescentus* S-layer (EMD-16183, PDB 8bqe) have now been submitted. We have also submitted the model for the new COP-II map (PDB-8bsh).

(8) Computational complexity is a very important topic in the context of SPT, but no details are included in the text other than the following statement: "Compared to M, RELION-4.0 uses computationally more efficient algorithms that do not require the computational power of a GPU". Does this imply that computationally inefficient algorithms require the use of GPUs? If the authors want to comment on computational complexity (which I think would be very important to do), actual numbers need to be presented, such as running times, compute resources used, storage, etc.

We did not imply this. We have rephrased the statement about GPU to:

“Compared to M, RELION-4.0 uses computationally more efficient algorithms; M uses GPUs to accelerate the calculations.”

To give better insights into the computational costs, we now include details of image processing requirements for all three data sets in Table 2.

(9) Regarding the use of a "true 2D approach" vs. "pseudo-volumes" (page 6), the following claim is made: "For example, the current implementation could be used to process pairs of tilted images, but a 2D implementation would be more efficient", does this refer to efficiency in computing terms, storage, or both? Also, the authors should comment on the storage requirements of their 3D approach. Since the pseudo-tomograms have three components (Eqs. 8-10), does this mean that they require 3x more storage compared to traditional subtomogram averaging? This would be an important practical consideration for prospective users of this tool and should be adequately discussed.

Yes, this is a good point, it would affect storage too, as the CTF^2^ and multiplicity volumes take up space as well. To explain this better, the revised text now reads:

“The use of 3D pseudo-subtomograms allowed re-using existing code for subtomogram averaging in RELION, while input stacks of 2D images would have required significant software development efforts. Nevertheless, in the future we might still choose to implement a true 2D version of the code, which would be more efficient, both in terms of processing time and disk storage requirements. In cases where the number of tilt images is small in comparison to the box size, fewer Fourier pixels need to be examined in a stack of 2D images than in a pseudosubtomogram, with a corresponding decrease in processing time. Moreover, the likelihood calculation from the 3D pseudo-subtomogram approach requires separate storage of the accumulated squares of the CTFs, and the corresponding multiplicity terms. In contrast, in the 2D approach, only the 2D images need to be stored, as CTF parameters can be calculated on-the-fly and there is no need for a multiplicity term, giving a corresponding decrease in storage requirements. However, if one were to collect tilt series with very fine angular increments or in a continuous manner, then the current implementation may still be preferable”.

10) In general, the use of the term "particle polishing" could lead to misinterpretations since it means different things in SPA and tomography. In SPA, "polishing" refers to the analysis of intermediate vidoe frames (which also exist in tomography), but the present study doesn't consider them at all. Also, I noticed that the term "frames" is used interchangeably to refer to SPA "vidoe frames" and "particle tilts" in tomography, which adds to the confusion. For example, the term "frame alignment" is mentioned twice on page 6, but I believe the authors are really referring to "tilt image alignment" instead. This should be clarified and the terms "frame" and "tilts" should be used consistently throughout the manuscript.

One complication here is that the terms “frame alignment” and “polishing” are already mentioned on the RELION-4.0 GUI and in its documentation. Still, we recognize the ambiguity of the word frame in our manuscript. Therefore, in all instances of the word “frame”, we explicitly refer to either “vidoe frame” or “tilt series frame” in the revised version.

That said, the actual program that performs the “frame alignment” can, in fact, also work with vidoe frames from the tilt series images. One problem here is that the regularization of the motions is not done separately for vidoe frames within one tilt image and vidoe frames of the other tilt images. This may explain why preliminary tests with the individual vidoe frames did not yield worthwhile improvements. In the end, we decided not to expose the vidoe-frame functionality to the end-user. We now explicitly mention in the Methods section:

“We also note that Bayesian Polishing in SPA describes particle motions between individual vidoe frames. Although our approach for tomography can also consider vidoe frames, the current implementation uses the same regularization of particle motions between vidoe frames within each tilt image as between the vidoe frames from other tilt images. Because preliminary tests showed limited benefits in considering the vidoe frames in this manner, only the functionality to model particle motions between the tilt series images were exposed on the GUI.”

11) For the *Caulobacter crescentus* S-layer reconstruction, the new cryo-ET map corresponding to the inner domain of the S-layer seems to have even better resolution than the SPA reconstruction reported in the original work [26] (3.5A vs. 3.7A for EMD-10389). Since the authors say these maps are in "excellent agreement", a quantitative comparison should be included. Was the second LPS binding site visible in the 3.7A SPA map? In Figure 2d, the fit between LPS and the cryo-ET density (right-hand side) seems to be of lower quality (i.e., no density for parts of the LPS model is seen). Without having access to the maps, it is not possible to properly evaluate these results.

All maps have now been submitted to the EMDB (also see point 7) and we have now included a comparison with the previous 3.7A map in Figure 2 —figure supplement 1. Please note that the 3.7 Å cryo-EM single-particle map was produced from data collected of an in vitro reconstituted specimen, rather than from the S-layer on native membranes on cells. The second LPS binding site was never observed in vitro, illustrating why in situ imaging is vital for obtaining physiologically relevant insights.

12) The paper is well-written and clear, but the description of the new concepts and evaluation of the method is somewhat minimalistic. While these may not be absolutely necessary, they will facilitate better understanding by a broader user community.(i) A graph describing the workflow as implemented in RELION, e.g., as a flow chart, would increase the readability of the paper.(ii) A figure visually describing the pseudosubtomograms would make the concept clearer to the reader.(iii) A more in-depth description of the input files (alignment files from imod, particle lists, and orientation from Dynamo/elsewhere) needed to perform alignments and averaging in RELION using pseudosubtomograms would be beneficial to the community. This could be incorporated into a flowchart as suggested in (i).(13) Along the same lines, the authors do not include any description of the particle-picking procedures or the software requirements. This seems to be important as the authors state "(particles) relative defoci are known from the geometry of the tilt series and the known 3D positions of the particles in the tomogram".

This paper describes the 3D pseudo-subtomogram approach with its correspondingly modified refinement algorithm, as well as the new functionality to perform CTF refinement and tilt series frame alignment in RELION-4.0. Unfortunately, RELION-4.0 does not yet provide a smooth pipeline for the entire processing procedure that would be captured well in a flow chart. Much improved import procedures and a smoother pipeline are the topic on ongoing work, which will be implemented in RELION-4.1. We have modified the Discussion to reflect this more clearly:

“However, tilt series alignment, tomogram reconstruction and particle picking are not yet part of the RELION workflow. Efforts to also implement solutions for those steps in a single tomography processing pipeline are ongoing and will be part of future RELION releases. Meanwhile, current import procedures rely on specific pre-processing operations in IMOD kremer1996computer, and particle coordinate conversion tools to use in RELION-4.0 are available for a range of third-party software packages pyle2022picking.”

(14) In section 2.2 it is shortly discussed that rotational priors can be used to constrain orientational searches. As this is often very useful for particle alignment in cryo-ET, this and the implementation should be described in more detail.

Performing local angular searches is done using RELION’s standard Gaussian-shaped priors on the Euler angles, which have been described previously. A reference to the corresponding paper has been inserted. Please also see our response to point (2).

(15) The progress in map quality is only shown by improvements to resolution in the FSC for the test datasets. It would be beneficial to also show the initial and intermediary maps for at least one of the test cases so that the reader can better assess the improvement in map quality. Overall maps should be shown to evaluate performance on the "whole particle" level, given considerations of flexibility (especially in cases where focused refinement was used). Corresponding local resolution maps would be informative in such cases.

We have inserted maps describing the stepwise improvement for the HIV immature capsid data set in Figure 1 —figure supplement 1.

(16) For the immature HIV capsid data, the FSC curves are only shown for the small subset of data. As it is an important point that the RELION pseudosubtomogram approach can reach the same resolution as previous software such as M, the data should also be shown for the full dataset (authors write "data not shown").

This has been done. See point (3).

(17) In general, a proper comparison to the previous approaches/results, maybe by providing the maps for comparison in the figures, would be helpful for the reader to evaluate which approach to choose, and based on what expectations/biological system.

See points (3), (4) and (11).

(18) In section 4 it is briefly discussed that particle rotation is not modeled in the per-particle motion correction. They draw parallels to single particles and conclude that this might be an issue at resolutions better than 2 Å. However, could the authors comment on whether the difference in dose and time frame of data collection as well as the thickness of sample, may affect the assumption that this is only an issue at a higher resolution?

The lower dose in tilt series images, and thicker samples will only make it more difficult to also model the rotations, as both will decrease signal-to-noise ratios in the images. The only advantage of tomographic data is that the distinct tilts provide complementary views of the same structure, which might benefit the rotational alignments.

(19) Many subtomogram averages are resolved at a more moderate resolution (e.g. 8-20 Å) than the maps presented in this study due to different reasons. It would be beneficial for the reader if the authors could comment on/discuss which optical and geometrical parameters can be confidently modeled at a lower resolution and if they improve resolution.

One would expect that shifts of the particles (during subtomogram alignment, but also during tilt series frame alignment) would still be reasonably well defined by such low-medium resolutions. Probably defocus refinement would be more difficult.

(20) Could the authors comment on whether it is possible to extract the refined optical and geometrical parameters for a set of particles from a tilt series, and apply them to a different set of particles from the same tilt series (or to the full tilt series to obtain a better quality tomogram for new particle picking).

This is not yet possible, but we have plans to implement this functionality in the future.

21) Along the same line: would it be possible to refine two distinct species (e.g. two distinct conformations of the same protein or two completely unrelated proteins) together in a multi-species refinement of optical and geometrical parameters? As the overall tilt series alignment and imaging parameters are the same, this might aid in refinement for lower abundance.

Not yet; also see point (19).

(22) On a more conceptual level, the reasoning to separate the SPA/Cryo-ET workflows completely in the software package remains unclear to me and it would be valuable if the authors could briefly discuss this.

The workflows are not separated completely. In fact, their implementation in RELION-4.0 is very much intertwined. This will aid newcomers to the field of tomography, who may already be familiar with SPA processing in RELION. The paper already makes this point.

23) Can the authors comment on the "The data model assumes independent Gaussian noise on the Fourier components of the cryo-EM images of individual particles" with respect to the more complex in situ data, where noise is expected to be more structured?

In neither SPA, nor tomography, is this assumption perfect. In both modalities, neighboring particles will introduce noise that is localized in real-space, and for which correlations thus exist in Fourier space. However, taking such correlations into account would be computationally difficult and expensive.